# Predictions of Land Use/Land Cover Change and Landscape Pattern Analysis in the Lower Reaches of the Tarim River, China

**Shanshan Wang** [1,2,3,4], **Qiting Zuo** [5,*] , **Kefa Zhou** [6], **Jinlin Wang** [1,2,3,4] and **Wei Wang** [1,2,3]

1. State Key Laboratory of Desert and Oasis Ecology, Key Laboratory of Ecological Safety and Sustainable Development in Arid Lands, Xinjiang Institute of Ecology and Geography, Chinese Academy of Sciences, Urumqi 830011, China; wangshanshan@ms.xjb.ac.cn (S.W.); wangjinlin@ms.xjb.ac.cn (J.W.); wangw1114@ms.xjb.ac.cn (W.W.)
2. Xinjiang Key Laboratory of Mineral Resources and Digital Geology, Urumqi 830011, China
3. Xinjiang Research Centre for Mineral Resources, Chinese Academy of Sciences, Urumqi 830011, China
4. University of Chinese Academy of Sciences, Beijing 100049, China
5. School of Water Conservancy Engineering, Zhengzhou University, Zhengzhou 450001, China
6. Technology and Engineering Center for Space Utilization, Chinese Academy of Sciences, Beijing 100049, China; zhoukf@ms.xjb.ac.cn
*   Correspondence: zuoqt@zzu.edu.cn

**Abstract:** Natural vegetation on both sides of the Tarim River Basin (TRB) is the only barrier—a critical ecological niche—between the economic belt in the artificial oasis and the Taklimakan Desert. To understand the impact of human activities on the TRB, we explored the spatial and temporal variations in land use/land cover change (LUCC) and landscape pattern evolution from 2000 to 2020. These variations were simulated for 2030 with the 20 years of data using the cellular automata–Markov model and geographical information system analyses. The results predicted substantial LUCCs in the lower reaches of the Tarim River (TRlr), with 3400 km$^2$ (20.29%) of the total area (16,760.94 km$^2$) undergoing changes. Wetland, artificial land, grassland, farmland, and forestland areas increased by 578.59, 43.90, 339.90, 201.62, and 536.11 km$^2$, respectively, during the period from 2020 to 2030. The only decreases were in the Gobi/other deserts and bare soils (1700.13 km$^2$). We also determined current and future changes in TRlr landscape pattern indices at the class and landscape levels. Combined with a field survey and hydrological data, theoretical support for effective land use management strategies is provided. The findings offer a scientific basis for future ecological civilization construction and sustainable development in the TRB.

**Keywords:** land use/land cover change; landscape pattern index; predictive analysis; human–land relationship; Tarim River





## 1. Introduction

The Tarim River Basin (TRB) is in the resource-dense area of southern Xinjiang Uygur Autonomous Region, China. It is an important strategic channel that directly connects Qinghai and Xinjiang and was an important stop along the ancient Silk Road [1,2]. The TRB plays a vital role in fostering economic development and expansion throughout Central and South Asia while also serving as a strategic energy and resource reserve base for China's continued progress [2]. Accordingly, ensuring the ecological health of forestland along the corridor represents the foundation for maintaining the regional ecological security of the river and ensuring the sustainable development of the oases [3].

Exacerbation of resource limitations, pollution, and ecological deterioration poses a direct threat to the sustainability of local economies, societal structures, and environmental security. Therefore, it is imperative to devise strategies aimed at ameliorating the vulnerability of the TRB environment [4,5]. In 2000, to restore and protect the downstream environment, the Chinese government implemented a comprehensive management system for the TRB [6,7]. To strengthen the management and maintenance of ecologically

fragile areas, it is important to scientifically assess, predict, and maintain their ecological security [1,3].

Land use/land cover changes (LUCCs) directly reflect the human activities that are carried out within natural environments [8,9]. Notably, land use characteristics display significant spatiotemporal characteristics, and time-series data obtained through remote sensing can provide a foundation for information acquisition and the prediction of future developmental trends [10,11]. The rapid development of landscape ecology fields related to LUCC research arose from the integration of modern disciplines, leading to the widespread application of landscape indices capable of quantitatively analyzing and describing LUCC patterns [12,13].

Advancements in remote sensing technology have prompted numerous researchers to propose a range of techniques for monitoring LUCCs and landscape patterns. For instance, El-Tantawi et al. [14] monitored the LUCCs in the western TRB during the period from 1990 to 2030 using an artificial neural network (ANN). Das et al. [15] evaluated the spatiotemporal characteristics of ecosystem health in the Kolkata Metropolitan Area by utilizing the normalized difference vegetation index (NDVI), landscape matrices, and LUCC data from 2000 to 2019. Zhang et al. [16] used the optimal parameter geographic detector (OPGD) model to quantitatively analyze a range of environmental indicators, including the NDVI, wetness (WET), normalized differential build-up and bare soil index (NDBSI), and land surface temperature (LST). They then integrated these indicators to construct a comprehensive index for evaluating the ecological environment. Siddik et al. [17] analyzed the relationship between LUCC alterations and groundwater recharge fluctuations in north-western Bangladesh by assessing the differences between wet and dry seasons and evaluating the temporal modifications in the period from 2006 to 2016. Li and Geng [18] used three-phase remote sensing image data of 2000, 2010, and 2020 to investigate the spatially positive correlations between the spatial aggregation effect and spatial and temporal changes in land use in the Guizhou Plateau Karst Watershed. Furthermore, Mu and Shen [19] conducted a dynamic assessment of landscape ecological risk within the TRB, revealing an optimized ecological spatial structure of arid inland rivers, which included the construction of a multicenter landscape ecological security pattern for the watershed. Moreover, Song et al. [20] analyzed the evolution characteristics of landscape patterns in Wuhan wetlands using remote sensing image interpretation data spanning from 2000 to 2020 and computing their landscape pattern index. In addition to these studies, research activities related to prediction and simulation have been carried out. Hou et al. [21] explored the contribution of the transition between different land types to the ecological environment based on LUCCs in the TRB during the period from 1990 to 2019 and those predicted in the next 10 years.

The above studies provide significant references for the study of landscapes and ecological security, including studies on LUCCs, the NDVI, the landscape pattern index, and some comprehensive indicators. However, previous studies have mainly focused on LUCCs, landscape ecology, or both, but have lacked comprehensive quantitative analysis and visualization of the dynamic evaluation of LUCC trends and landscape pattern changes, as well as predictive simulations at different spatial and temporal scales.

The present study used the theory of landscape ecology, remote sensing information processing, the CA–Markov model, GIS, and geostatistical methods to investigate the lower reaches of the Tarim River. We aimed to: (i) obtain the LUCC classification diagrams of the lower reaches of the Tarim River (TRlr) using Landsat satellite imagery from 2000, 2010, and 2020, along with a digital elevation model (DEM), slope, buffer zone, and similar data, which were used to simulate the 2030 land use data based on the cellular automata–Markov (CA–Markov) model; (ii) assess the current and future changes in the overall regional LUCC; and (iii) examine the spatiotemporal trends of landscape pattern changes in the TRlr during the period from 2000 to 2030. Combined with a field survey and hydrological data, the findings provide an in-depth understanding of the human–land relationship and ecological trends under societal influence, theoretical support for formulating effective

ecological land use management strategies, and evaluation techniques and methods for the sustainable development of the TRB.

## 2. Study Area

The Tarim River mainstream forms in Xiaojiake at the confluence of the Akesu, Yarkant, and Hotan Rivers and empties into Taitema Lake. It has a total length of 1,321 km, which can be divided into the upper, middle, and lower reaches [22]. The ecological environment of this region is extremely fragile [6,23], with an average annual precipitation of only ~51.2 mm and an evaporation demand of 2300–3000 mm. The local water resources include melt water, forest precipitation, and bedrock fissure waters in the high, middle, and low mountainous areas, respectively. However, the water supplies to the Tarim River generated by the latter two are weak because of its location deep within the continent (far from the ocean) surrounded by high mountains. Accordingly, snowmelt from the surrounding alpine glaciers is the main water source, which generates near-zero flow after it exits the mountains.

The study area is a dry-flow area in the TRlr located within an altitude range of 801.50–846.25 m, with the Taitema Lake area representing the lowest point (801.50 m). The length of the TRlr (Qara to Taitema Lake) is 428 km (Figure 1). Ecological problems in this region are becoming prominent owing to extended periods of river drying. Land use problems also manifest as expanding artificial oases in the middle and upper reaches and shrinking natural oases in the lower reaches. Further issues related to deforestation and abandoned farmlands are present in the upper and lower reaches, respectively [6,7]. Moreover, large amounts of water have been diverted to the irrigation areas of the oases in the upper and middle reaches, resulting in an uneven distribution of water resources, rising groundwater levels, and secondary salinization in the region [24–26]. Intermittent ecological water transfers were implemented in 2000 to restore the ecological conditions of the TRlr, which altered the ecology and environment of the dry-flow areas and resulted in corresponding changes in the LUCC and landscape pattern characteristics [27,28].

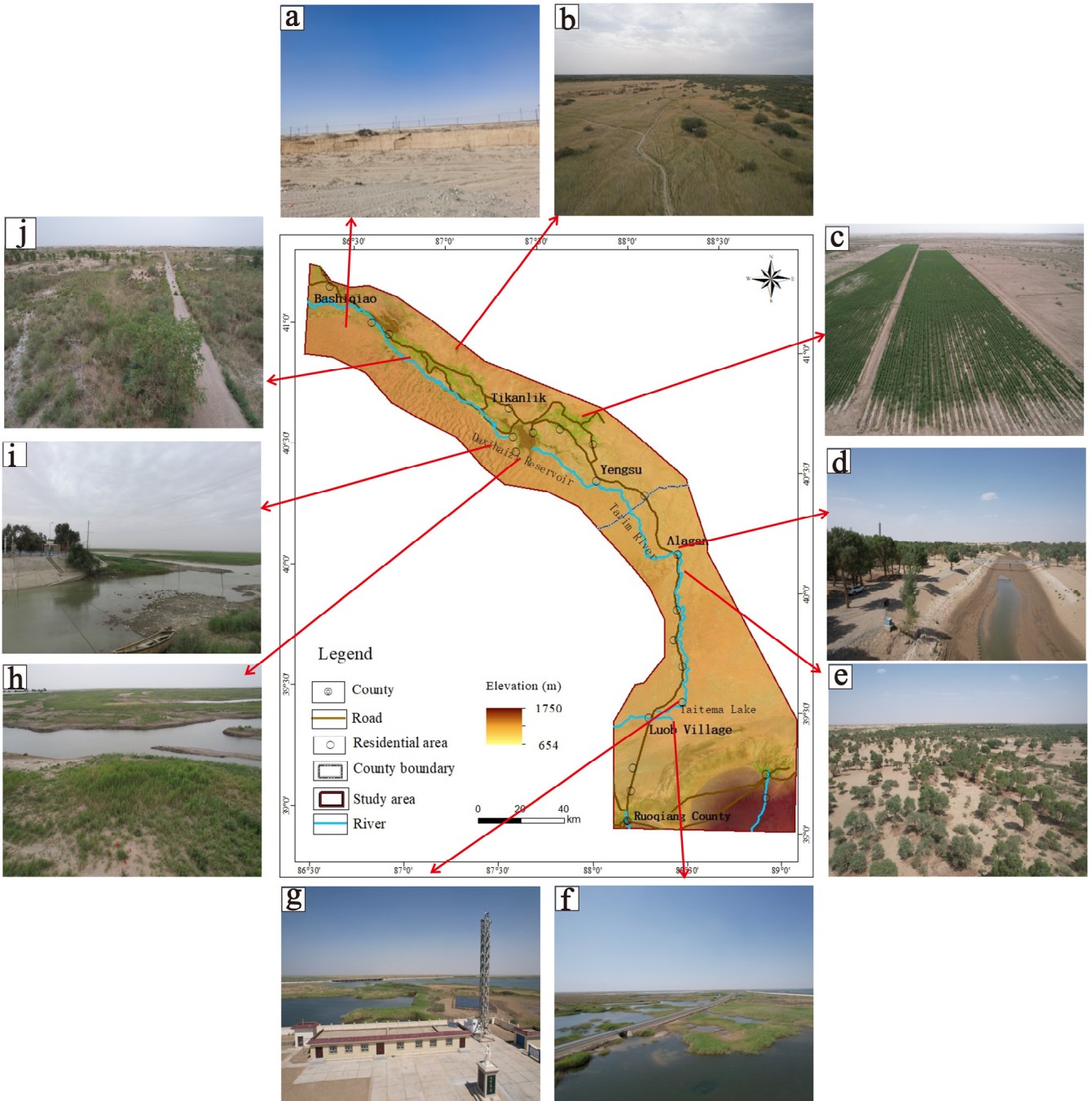

**Figure 1.** Schematic overview of the study area and the location of the studied landscapes of: (**a**) other land; (**b**) grassland; (**c**) farmland; (**d**) wetland; (**e**) forestland; (**f**) wetland; (**g**) artificial land; (**h**) grassland and wetland; (**i**) artificial land and wetland; (**j**) grassland and forestland in the lower reaches of the Tarim River (TRlr).

## 3. Materials and Methods

### 3.1. Data Sources

The collection of remote sensing imagery data for this study centered on the TRlr, including Taitema Lake. A digital elevation model (DEM) of the study area was acquired from the geospatial data cloud (https://www.gscloud.cn) (accessed on 10 November 2021) at a 30 m spatial resolution. The slope data were extracted in ArcGIS 10.2, and soil, water source, vegetation type, road, and protected area zone data were provided by the departments of the TRB overseeing the respective basin areas. TRlr satellite imagery for 2000,

2010, and 2020 were obtained from Landsat Thematic Mapper (TM), Landsat Enhanced Thematic Mapper Plus (ETM+), and Operational Land Imager (OLI) data, which maintain a 30 m resolution. Subsequently, land use types in the study area were categorized with the support of ENVI.5.3 and ArcGIS, and characteristics of land resources and requirements were extracted according to existing land use classification principles for accuracy analyses. Six level-1 land use units were derived, namely grassland, forestland, wetland, farmland, artificial land, and other land.

The 2000 and 2010 Landsat images were corrected geometrically using the quadratic polynomial in the image-to-image registration method, with the 2020 images as the base. The main ground control points (GCPs) were roads, river intersections, building boundaries, and isolated vegetation points far from roads. At a minimum, 30 evenly distributed GCPs per plot were used, with the density of the GCP distributions increasing along the riverbanks. To meet the accuracy requirements when preparing the images, the residual error was ensured to be within one pixel during registration.

Nine remote sensing images were inlaid to facilitate the preparation of analysis imagery. One image occupying the largest proportion of the study area with a good data quality was selected as the reference image during the inlay process, whereby the colors of the remaining images were normalized based on this reference image. Subsequently, the FLAASH (Fast Line-of-sight Atmospheric Analysis of Spectral Hypercubes) module in ENVI was used for atmospheric corrections between two images.

### 3.2. CA–Markov Model Construction

A cellular automaton (CA) represents a network dynamics model comprising cells with various spatiotemporal characteristics and domains dependent on intra-cellular and neighborhood interactions [21,29]. Notably, this model can simulate the spatiotemporal evolution of complex systems. Comparatively, the Markov model is based on the Markov stochastic process and describes the changing characteristics of an event over a particular period. This information is used as the basis to predict the future development trends of an event. Here, the LUCCs of the study area were simulated using ArcGIS, and the CA–Markov module was implemented in IDRISI v.-17.0. The accuracy of the 2020 simulation model was tested prior to simulating the TRlr 2030 LUCC imagery using the processes described below (Figure 2).

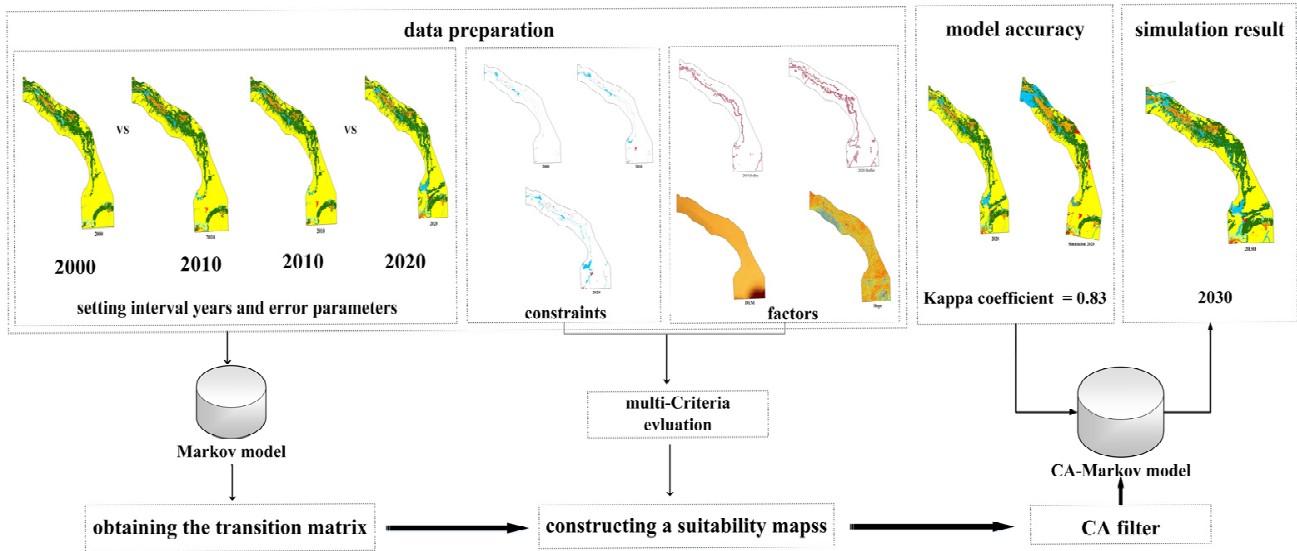

**Figure 2.** Flowchart of research methods. CA—cellular automaton.

- Step 1—Data preparation. Land use data mosaicking and clipping, as well as coordinate system and resolution adjustments, were carried out in ArcGIS. Before the

2000–2010 LUCC images were imported, all raster data were converted into a format recognizable by IDRISI. These image files, combined with the DEM, slope, and other data, were converted to ASCII format for reclassification.

- Step 2—Obtaining the transition matrix. CA land use simulations are advantageous for generating transition potential diagrams that consider spatial structures and neighborhood status. Markov chain analyses can provide a matrix for LUCC transition zones based on temporal variations. The 2000–2010 transition matrix and conversion probability of land use types were obtained using the Markov module considering the base year land use data, in addition to setting the error parameters and interval years.
- Step 3—Constructing a suitability atlas. The transition matrix and conversion probability of land use types were applied as computational rules in the CA–Markov model. The suitability of various land use types during the evolutionary process and the effects between cell neighborhoods were comprehensively considered. Artificial land and wetland were set as restricted land use types. Buffer zones of various distances (50, 100, 200, and 500 m) were designated as influencing factors along wetland peripheries. The overall purpose was to conduct an in-depth analysis of land use variation impacts within the buffer zones based on model predictions. Lastly, the adaptive images of the various land use types were compiled into an atlas.
- Step 4—CA filter and number of cycles. The number of CA cycles in the model was set to 10 using the CA–Markov module. Subsequently, a $5 \times 5$ m neighborhood filter was used to predict the land use status in 2020.
- Step 5—Model accuracy. The *CROSSTAB* tool in IDRISI was used to calculate the kappa coefficient (0.83). Because the kappa coefficient was >0.75, the simulation results could be regarded as highly consistent with the ground truth data, and thus to have good reliability.

The above steps were repeated using the 2010–2020 data. Subsequently, the same number of cycles and neighbourhood filters were set to simulate the predicted 2030 LUCC diagrams.

### 3.3. Landscape Pattern Index

The landscape pattern index quantitatively expresses the changing characteristics of the landscape pattern while reflecting landscape fragmentation, proximity, diversity, and convergence [30–32]. ArcMap was used to convert the vector map of land uses into a raster layer with a precision level of $30 \times 30$ m before the latter was input into FRAGSTATS v.4.2. Nine landscape indices were then selected to accurately reflect the situation of the TRlr and according to the research needs, namely number of patches (NP), patch density (PD), and mean patch size (AREA_MN) at the type level, and Shannon diversity index (SHDI), Shannon evenness index (SHEI), largest patch index (LPI), landscape shape index (LSI), landscape division index (DIVISION), and landscape contagion index (CONTAG) at the landscape level. All landscape pattern indices and their ecological significances are shown in Table 1.

**Table 1.** Landscape pattern indices and their ecological significance.

| Index Type | Index Name | Abbreviation | Level | Ecological Significance |
|---|---|---|---|---|
| Area index | Mean patch size | AREA_MN | C/L | Degree of contagion or division of each patch type in the landscape |
| | Largest patch index (proportion of overall patch) | LPI | C/L | Dominant landscape type and level of human disturbance |
| Vergence index | Patch density | PD | C/L | Degree of contagion and division within a particular landscape type |
| | Number of patches | NP | C/L | Complexity of the landscape spatial structure |
| | Landscape contagion index | CONTAG | L | Contagion and division tendencies of different patch types in the landscape |
| | Landscape division index | DIVISION | C/L | Distance between patches |

Table 1. *Cont.*

| Index Type | Index Name | Abbreviation | Level | Ecological Significance |
|---|---|---|---|---|
| Shape index | Landscape shape index | LSI | C/L | Migration characteristics of species and energy flow in landscape pattern |
| Diversity index | Shannon diversity index | SHDI | L | Abundance of landscape |
| | Shannon evenness index | SHEI | L | Proportion of landscape affected by the dominant patch type |

Notes: C = class level; L = landscape level.

## 4. Results

### *4.1. Simulations and Predictions of TRlr LUCCs*

#### 4.1.1. Predicted 2030 LUCC Diagram for TRlr

The model predicted the changes in the TRlr landscape patterns in 2030 (Figure 3) and showed that other land (unused) would become the predominant land use type in the region, with an area of 7548.56 km$^2$ (accounting for 45.04% of the total area). Forests were predicted to represent the second largest land use type and were distributed widely across all reaches of the Tarim River, accounting for ~33.53% (5619.50 km$^2$) of the total area. Wetland was the third largest predicted land use type, predominantly comprising rivers, reservoirs, and lakes, and covering an area of 1358.34 km$^2$ (accounting for 8.10% of the total area). Notably, the wetland area in the TRlr was predicted to increase in 2030. Farmland and grassland were the next largest in size, covering areas of 1066.88 km$^2$ (6.37%) and 1038.21 km$^2$ (6.19%), respectively. Farmland, in both planar and banded forms, was distributed primarily in the upper and lower reaches, whereas grassland was concentrated in the lower reaches. Artificial land was dominated by mines and construction land, covering a total area of 129.45 km$^2$ (0.77%) scattered in the middle and lower reaches, indicating that the predicted artificial land area increased more than in the years of 2000, 2010 and 2020.

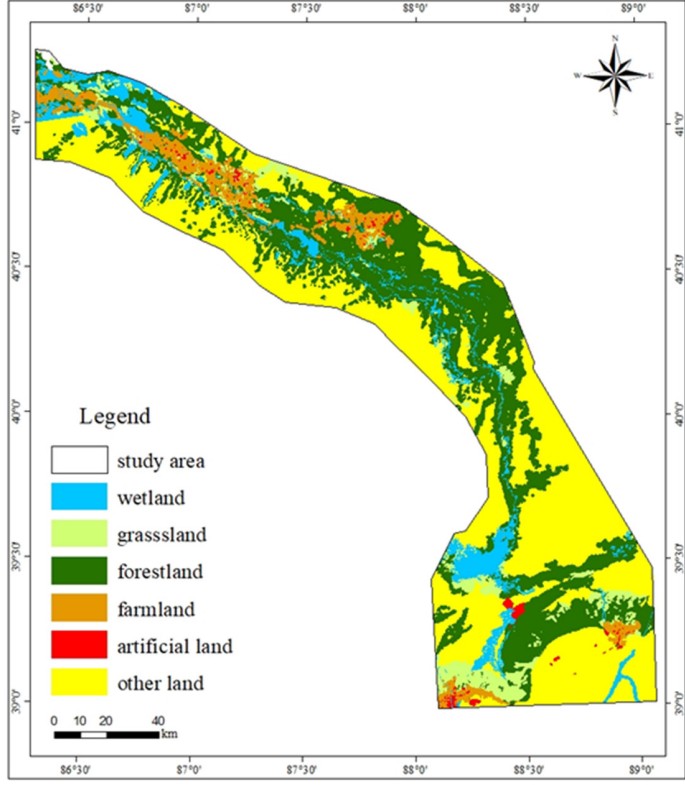

**Figure 3.** Predicted 2030 land use/land cover change (LUCC) map of the lower reaches of the Tarim River (TRlr).

### 4.1.2. Analysis of Predicted Land Use Trends

The TRlr LUCC transition matrixes for 2020 and 2030 were calculated for further analysis (Table 2). The results predicted substantial LUCCs in the TRlr, with 3400 km$^2$ (20.29%) of the total area (16,760.94 km$^2$) undergoing changes. Wetland, artificial land, grassland, farmland, and forestland areas increased by 578.59, 43.90, 339.90, 201.62, and 536.11 km$^2$ (74.20, 51.31, 48.68, 23.30, and 10.55%), respectively, whereas only the areas of the Gobi/other deserts and bare soils (both classified as other land) decreased (by 1700.13 km$^2$, i.e., 18.38%).

**Table 2.** LUCC transition matrix of the lower reaches of the Tarim River (TRlr) for 2020–2030 (km$^2$).

|  | Grassland | Farmland | Forestland | Other Land | Artificial Land | Wetland | Total in 2020 |
|---|---|---|---|---|---|---|---|
| **Grassland** | 686.07 | 7.56 | 0.18 | 0.00 | 3.33 | 1.16 | 698.30 |
| **Farmland** | 2.14 | 841.89 | 0.03 | 0.02 | 19.59 | 1.60 | 865.26 |
| **Forestland** | 120.86 | 195.14 | 4602.30 | 0.26 | 2.43 | 162.41 | 5083.39 |
| **Other land** | 227.23 | 20.39 | 1015.55 | 7548.27 | 22.26 | 414.99 | 9248.69 |
| **Artificial land** | 0.05 | 0.72 | 0.08 | 0.01 | 81.81 | 2.90 | 85.55 |
| **Wetland** | 1.85 | 1.19 | 1.37 | 0.00 | 0.04 | 775.29 | 779.74 |
| **Total in 2030** | 1038.21 | 1066.88 | 5619.50 | 7548.56 | 129.45 | 1358.34 | 16,760.94 |
| **Area of change** | 339.90 | 201.62 | 536.11 | −1700.13 | 43.90 | 578.59 | 0.00 |
| **Rate of change (%)** | 48.68 | 23.30 | 10.55 | −18.38 | 51.31 | 74.20 | 0.00 |

Other land was predicted to be converted into wetland, artificial land, grassland, farmland, and forestland. The wetland area increased by 578.59 km$^2$ (74.20%) because of conversion of other land (71.72%) and forestland (28.07%). The rapid expansion (51.31%) of artificial land—such as construction lands, transportation lands, and mines—was predicted to result from the conversion of other land (50.70%) and farmland (44.62%). The increase in grassland area by 339.90 km$^2$ (48.68%) was ascribed to the conversion of other land (66.85%) and forestland (35.56%). Comparatively, farmland increased by 23.30%, of which 96.78% was converted from forested land. The increase of 536.11 km$^2$ (10.55%) in forestland was entirely because of the conversion from other land, completely compensating for the forestland area converted into wetland, grassland, and farmland.

### 4.2. Spatiotemporal Characteristics of LUCCs

The classification results of the TRlr LUCCs by land use types in 2000, 2010, 2020, and 2030 (predicted) were obtained following the interpretation of remote sensing data, and predictions based on these data. Subsequently, the LUCC spatial distribution characteristics and the corresponding land area of the changes were determined through statistical analyses. Both banks of the TRlr are covered predominantly by forests, with sporadic grassland and farmland (Figure 4). Some wetlands are also present, such as lakes, reservoirs, and ponds. Artificial land at the farmland peripheries is used for construction, transportation, and mining. The composition of the various LUCC types is shown in Figure 5.

The LUCCs for 2000–2020 and 2030 (predicted) are compared in Figure 5. Forestland occupied the largest area (5619.50 km$^2$) under the predicted situation in 2030, and the second largest area in 2020 (5083.45 km$^2$). The grassland area was predicted to be the second largest in 2030 (1038.21 km$^2$); however, it was the smallest in 2010 (493.09 km$^2$). The farmland areas increased continuously in 2000, 2010, and 2020 and were predicted to increase further by 2030 (517.11, 690.67, 865.27, and 1066.88 km$^2$, respectively). The predicted area of farmland in 2030 increased by 549.77 km$^2$, i.e., more than double that in 2000. The magnitude of the increase in the wetland area was the greatest, whereby the predicted area in 2030 increased by 1138.05 km$^2$ (more than 6-fold) compared to that in 2000. The artificial land area was predicted to increase from 34.72 km$^2$ in 2000 to 129.45 km$^2$ in 2030, representing a 272.83% increase. The area of other land (Gobi/other deserts and

bare soils) was predicted to decrease by 2834.92 km² in 2030 compared to that in 2000, representing a 27.30% reduction.

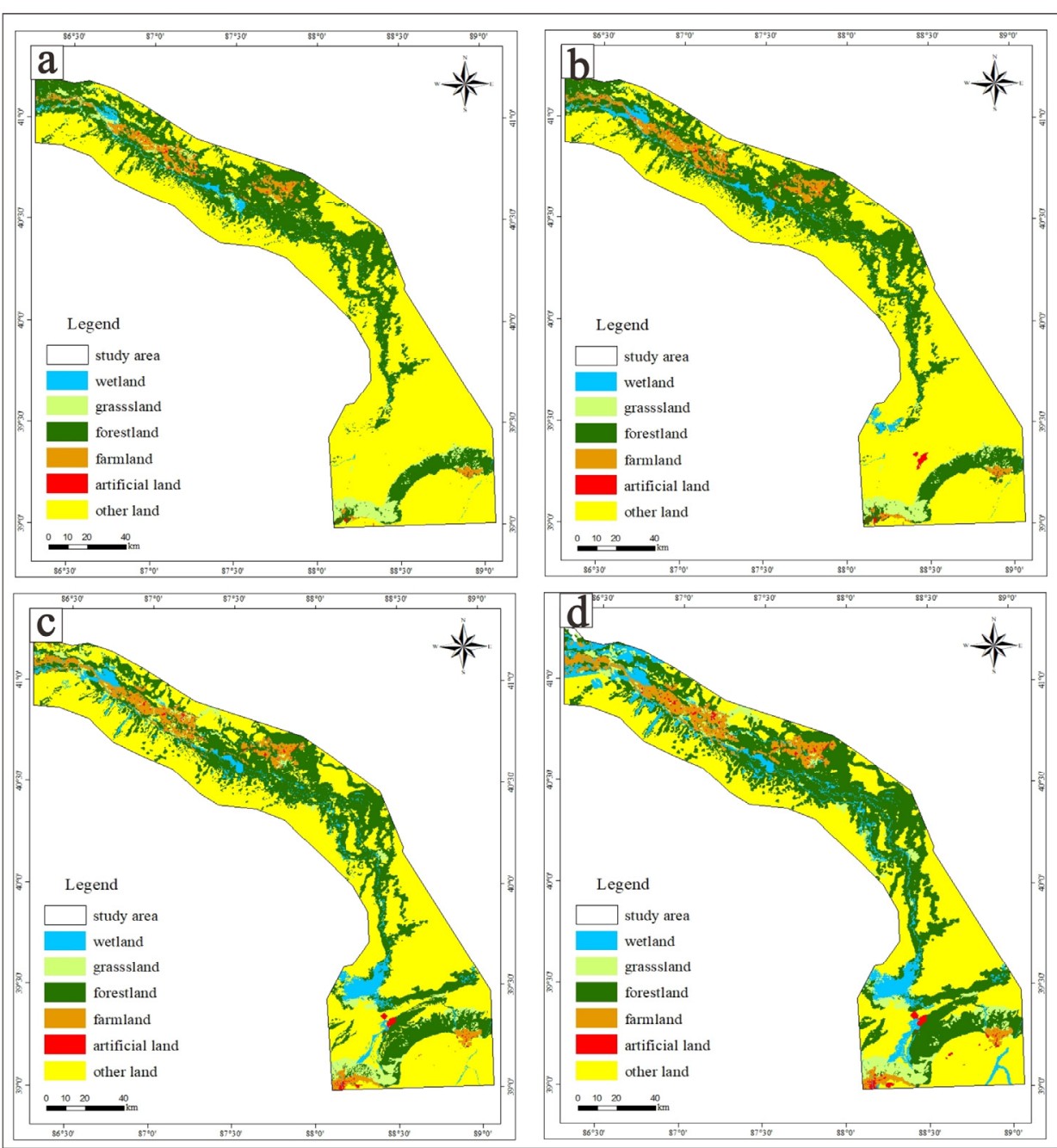

**Figure 4.** Land use/land cover change (LUCC) classification diagrams of the study area over four time periods: (**a**) 2000, (**b**) 2010, (**c**) 2020, and (**d**) 2030.

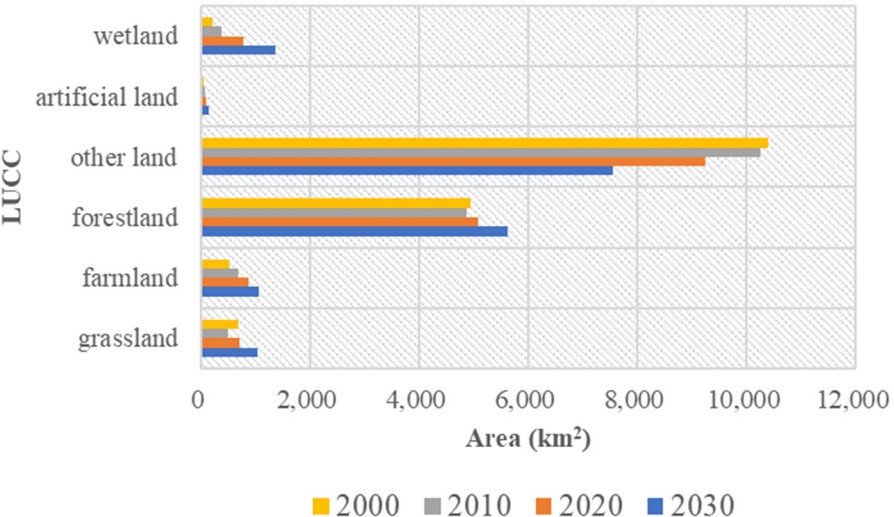

**Figure 5.** Changes in the land area of various land use/land cover change (LUCC) types over four time periods.

*4.3. Evolutionary Characteristics of TRlr Landscape Pattern Indices*

4.3.1. Class-level Analysis of Landscape Pattern Indices

The TRlr landscape pattern changes at the class level are shown in Table 3. The NP of other land was the largest, and that of the remaining land uses decreased in the following order: grassland > forest > wetland > farmland > artificial land. The PD of other land was much higher than that of the remaining land use types, whose PD decreased in the following order: grassland > wetland > farmland > forestland. Comparatively, the PD of artificial land was the lowest. The AREA_MN decreased in the order of forestland > farmland > other land > artificial land > grassland > wetland.

**Table 3.** Values of landscape pattern indices of the lower reaches of the Tarim River (TRlr) at the land use type level. NP—number of patches; PD—patch density; AREA_MN—mean patch size.

| Index | Year | Wetland | Farmland | Artificial Land | Forestland | Grassland | Other Land |
|---|---|---|---|---|---|---|---|
| NP | 2000 | 620 | 1194 | 791 | 321 | 1281 | 542 |
| | 2010 | 523 | 259 | 241 | 558 | 388 | 316 |
| | 2020 | 471 | 249 | 267 | 434 | 152 | 164 |
| | 2030 | 525 | 371 | 195 | 1429 | 990 | 8109 |
| PD | 2000 | 0.04 | 0.07 | 0.05 | 0.0192 | 0.0764 | 0.0323 |
| | 2010 | 0.03 | 0.02 | 0.01 | 0.0333 | 0.0231 | 0.0189 |
| | 2020 | 0.03 | 0.01 | 0.02 | 0.0259 | 0.0091 | 0.0098 |
| | 2030 | 0.01 | 0.01 | 0.01 | 0.0243 | 0.0169 | 0.1382 |
| AREA_MN | 2000 | 108 | 43.1 | 624 | 3234.723 | 17.2144 | 6.3591 |
| | 2010 | 70.4 | 267 | 32.5 | 872.8395 | 127.1125 | 3246.939 |
| | 2020 | 166 | 3714 | 262 | 1171.325 | 569.327 | 52.1473 |
| | 2030 | 259 | 289 | 66.5 | 394.1325 | 105.2774 | 93.2205 |

Analyses of the trends in NP, PD, and AREA_MN for 2000–2020 and 2030 (predicted) showed a decrease in the number and density of various land use type patches. Except for that of other land, the AREA_MN of the remaining land use types increased, indicating a weakening degree of division and higher continuity. The AREA_MN of farmland, grassland,

and wetland continued to increase, indicating that the ecological landscape patterns became more aggregated and stable.

The predicted 2030 data showed that the NP of other land, forestland, grassland, wetland, and farmland increased compared to that in 2020. The AREA_MN of farmland, artificial land, forestland, and grassland decreased, indicating a greater degree of division over the 10-year predictive period. Nevertheless, the changes were relatively stable compared to the multi-year changes observed since 2000.

4.3.2. Landscape-Level Analysis of Landscape Pattern Indices

Landscape pattern indices were used to analyze the study area and reveal the characteristic variations between patches. The convergence, diversity, and shape index values are shown in Table 4.

The NP values were the highest in 2000 and 2010 and lower in 2020. Further, the degree of landscape division decreased in 2020. The 2020 LPI (18.81) was lower than that in 2000 and 2010, reflecting the dominant landscape type. Additionally, the human disturbance level in 2020 was lower than that in 2000 and 2010.

**Table 4.** Landscape pattern indices of the lower reaches of the Tarim River (TRlr) at the landscape level. LPI—largest patch index; LSI—landscape shape index; CONTAG—landscape contagion index; DIVISION—landscape division index; SHDI—Shannon diversity index; SHEI—Shannon evenness index.

| Year | NP | LPI | LSI | CONTAG | DIVISION | SHDI | SHEI |
| --- | --- | --- | --- | --- | --- | --- | --- |
| 2000 | 4749 | 45.56 | 28.72 | 71.08 | 0.73 | 0.96 | 0.54 |
| 2010 | 2285 | 44.97 | 27.35 | 70.00 | 0.75 | 1.00 | 0.56 |
| 2020 | 1737 | 18.81 | 33.48 | 65.59 | 0.91 | 1.15 | 0.64 |
| 2030 | 11,625 | 44.55 | 27.09 | 73.96 | 0.73 | 0.97 | 0.50 |

In addition to playing an important role in the study of landscape function, such as species migration and energy flow, LSI can be employed to reveal the landscape pattern characteristics. The LSIs of 2000, 2010, and 2020 were 28.72, 27.53, and 33.48, respectively. CONTAG refers to landscape contagion and is an index that shows whether the landscape components are aggregated or distributed across multiple elements. The CONTAG values are higher when a particular patch type maintains absolute dominance and aggregation. Here, the values for 2000 and 2010 were high, reaching the minimum (65.59) in 2020. The result indicated that the aggregation of any patch type with an advantage was lower in 2020 compared to the previous two decades (Figure 6a–c).

The DIVISION index reflects the distances between patches, landscape distribution, connectivity, and resistance to disturbances. The DIVISION values for 2000 and 2010 were similar, and the value for 2020 was the highest (0.91); thus, the more aggregated the various patches were in 2020, the higher the landscape contagion and the greater its ability to resist disturbances. The SHDI and SHEI reflect the diversity and evenness of the landscape, respectively, and had higher values in 2020 than in 2000 and 2010. This finding indicated that the patch types had increased over the last 20 years, and the number of landscape components had also increased. Here, the larger the SHEI values, the less severely that the landscape was affected by one or several dominant patches.

The NP value increased substantially in the forecasted landscape pattern indices for 2030, with the degree of landscape fragmentation predicted increasing in the 10-year period from 2020 to 2030, and the resulting landscape spatial structure becoming more complex and diversified. The LPI was also expected to increase more than the 2020 values, with increases in the dominant landscape type and human disturbance level. Comparatively, the predicted LSI in 2030 was lower than that in 2020 and more like that in 2000 and 2010.

The predicted CONTAG in 2030 was higher than that in 2020, indicating a stronger degree of aggregation and an extension of the dominant patch type in the landscape (Figure 6).

The predicted 2030 DIVISION value was lower than that in 2020, with an expected reduction in the intra-patch distance. This situation reflected a more fragmented landscape distribution, lower contiguity, and a weaker ability to resist disturbances. The predicted SHDI and SHEI values for 2030 were also lower than those in 2020, indicating that the abundance of the landscape is likely to decline over this 10-year period. Accordingly, the impacts of the dominant patch would be enhanced.

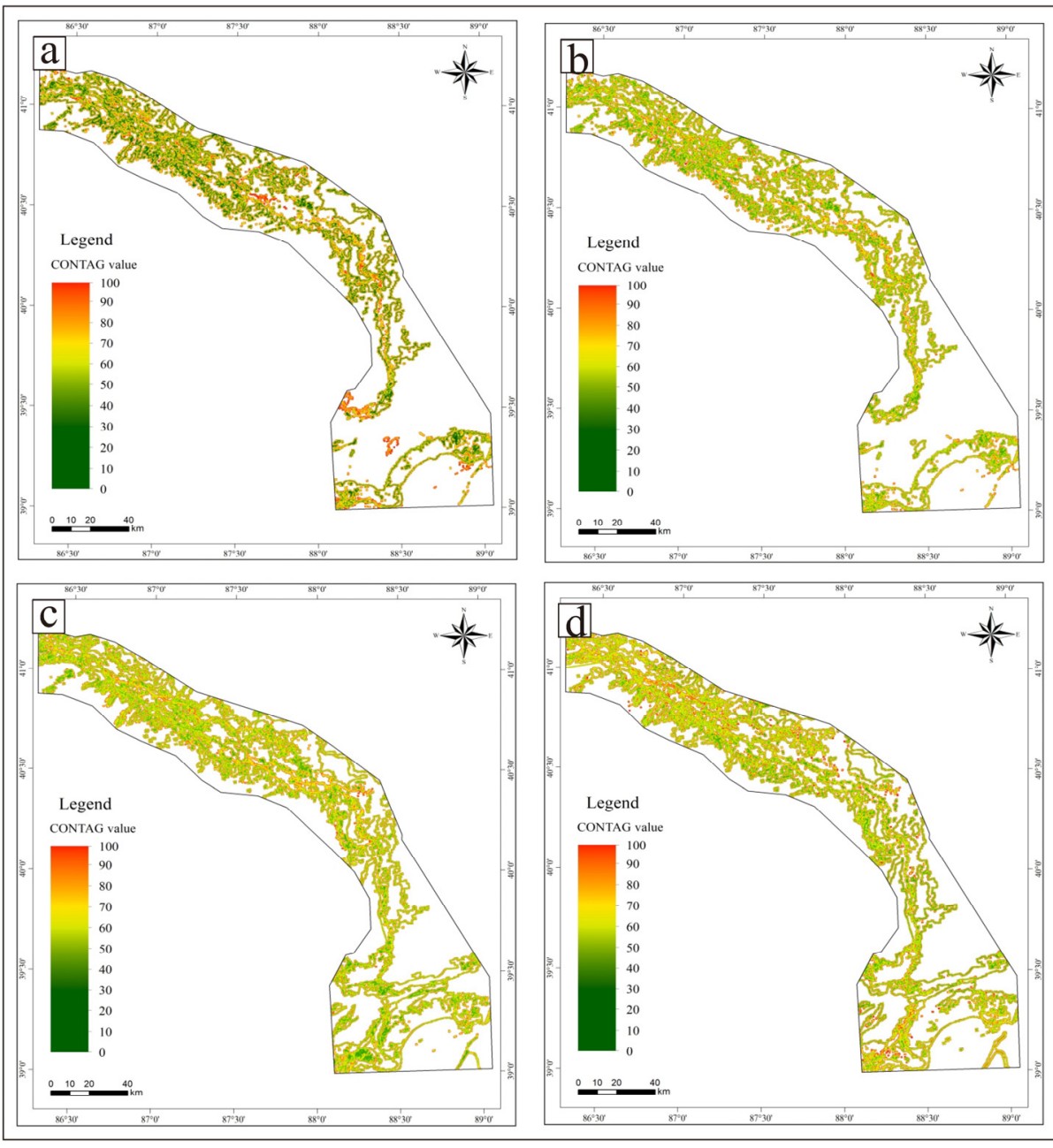

**Figure 6.** Landscape pattern indices (CONTAG) of the study area over four time periods: (**a**) 2000, (**b**) 2010, (**c**) 2020, and (**d**) 2030.

## 5. Discussion

### 5.1. Landscape Pattern Trends in the Period of 2000–2020

The People's Government of the Xinjiang Uygur Autonomous Region implemented the "Emergency Ecological Water Conveyance Project" in the TRlr in 2000. A total of 21 ecological water conveyances were completed in the following two decades, totaling ~8.43 billion m$^3$. The water conservancy head reached Taitema Lake numerous times to form a large waterbody area, and double-channel water delivery was realized on multiple occasions [19,33,34]. Long-term monitoring data indicated that the groundwater level in the TRlr within 1 km of the mainstream rose from the original 8–12 m to <4 m [26]. Consequently, the ecological benefits improved significantly [13,28]. Continuous water delivery ended the 30-year history of sustained interrupted flow in the TRlr and effectively alleviated the passive situation of severe ecological degradation in the TRB. Furthermore, such large-scale and intermittent artificial water delivery projects are rare, both in China and worldwide [1,13,23].

Throughout the period of 2000 to 2020, other land types—mainly Gobi/other deserts and bare soils—accounted for nearly 50% of the TRlr LUCC types; however, this land use type presented a decadal decrease. In each data period, the proportions of forestland were relatively high, accounting for 29.45%, 29.06%, and 30.33% in 2000, 2010, and 2020, respectively, followed by farmland, grassland, and wetland, whereas artificial land accounted for the lowest proportion. The number and density of patches of the various land use types decreased during the period from 2000 to 2020, whereas the AREA_MN increased for all land use types, except for other land. The area of AREA_MN exhibited a ranking order of forest > farmland > other > artificial > grassland > wetland, indicating that the overall ecological landscape pattern became more aggregated and stable.

The LUCC and landscape pattern changes for forestland, grassland, and wetland displayed an increasing trend during the period from 2000 to 2020; however, human–water system interactions across the TRlr constantly changed during this period. Notably, the development and utilization of water resources must be adapted to these changes, and specific measures should be taken to improve the coordination of the human–water relationship and ensure that these key point and region compositions continue to play a supporting role in socioeconomic development.

### 5.2. Predicted Landscape Pattern Trends

In this study, a CA–Markov model and GIS spatial analysis techniques were used as a novel method for extracting time-series LUCC information across various land use types and landscape pattern changes. Chinese and international researchers have conducted numerous empirical analyses of the evaluation and prediction of landscape ecological security, achieving realistic results [35]. Moreover, spatial analyses could be carried out with other related elements to establish a high-precision, intelligent, and universal data foundation to serve as a basis for large-scale reproduction and prediction of LUCC trajectories.

The total land area predicted to change amounted to ~3400 km$^2$, accounting for 20.29% of the total study area. Specifically, the wetland, artificial land, grassland, farmland, and forestland areas increased by 74.20, 51.31, 48.68, 23.30, and 10.55%, respectively. The only exception was the area of other land (including Gobi/other deserts and bare soils), which decreased by 18.38%. In general, other land was predicted to shift to more functional uses, indicating a robust developmental trend of LUCCs across the TRlr.

Comparatively, the landscape patterns in 2030 predicted that the NP of other land, forestland, grassland, wetland, and farmland would increase, whereas the mean patch sizes of farmland, artificial land, forestland, and grassland would decrease. Further, the degree of fragmentation was predicted to increase, with the landscape spatial structure becoming more complex and diversified. LPI would increase in 2030, indicating increases in the growth of the dominant land use type and the human disturbance level. On the other hand, LSI was predicted to decrease in 2030, whereas CONTAG values were predicted to increase compared to those in 2020. These predictions indicated a trend of increase

in the aggregation and extension of different patch types across the landscape in 2030. The decrease in DIVISION values suggested reductions in the distance between patches, manifested in terms of a more dispersed landscape distribution as well as reduced contagion and a weaker ability to resist disturbances. The SHDI and SHEI values were also predicted to decrease in 2030 compared to those in 2020. The SHDI reflected the abundance of the landscape and indicated a lower abundance of patch types in 2030. Alternatively, the lower SHEI values suggested that the landscape was more affected by one or several dominant patches [36,37].

Landscape abundance was predicted to decrease by 2030, whereas the influence of the most dominant patches in the landscape was predicted to increase. The predictions indicated that the TRlr landscape would suffer further from marked fragmentation and worsening degrees of patch isolation. This finding is consistent with other landscape patterns observed in the TRB [19,22,38]. Specifically, landscape pattern risks were predicted to increase between 2020 and 2030, which would be primarily manifested in the Gobi/other deserts, bare soils, and the Taitema Lake Basin located downstream, as well as the buffers and transition zones on both riverbanks.

### 5.3. Measures and Suggestions

The TRB should be regarded as a complex system at the interface of nature, human society, and economy. Accordingly, an overall perspective should be adopted to analyze the relationships and interactions between regional socioeconomics, natural ecology, and water resources within the basin, in addition to exploring the causes of ecological degradation within the area [39–41]. To this end, identifying the specific LUCC dynamics and patterns provides the key foundation for strengthening federal scientific supervision, promoting sustainable and socioeconomically beneficial land use, and alleviating human–land conflicts and other ecological issues [42–44]. After accounting for the implementation of water delivery and the existing problems in the TRlr, we propose the following suggestions:

1. According to our analysis of the spatiotemporal trends of LUCCs in the period of 2000–2030, attention should be paid to the key locations of ecological protection, and projects for comprehensive improvement of river channels should be implemented, mainly in the region of Daxihaizi Reservoir, Alagan, and Tikanlik. Certain land use types such as forestland, grassland, and wetland should be given more attention to promote the future expansion of the local vegetation area into the key locations.

2. Ecological water transmission should be scientific, efficient, and economical. The buffer zones at different distances from the central wetland (50, 100, 200, and 500 m) as influencing factors indicated that vegetation restoration occurs mainly in areas such as those near river channels, lakes, and ponds. Therefore, river construction maintenance and targeted water transmission methods maximize and optimize water transmission benefits and configuration [4,45]. To adjust the volume of river overflow, intensity of interference, and water transmission time, factors such as river overflow, linear water transmission, double river channels, and surface water transmission should be used to avoid the concentrated dissipation of river water in natural conditions.

3. The ecological monitoring network system should be improved by integrating the advanced Internet of Things, remote sensing, and big data technologies to achieve "space–ground integration". This system could theoretically allow the dynamic monitoring of surface water, groundwater, land use, natural vegetation, wild animals, desertification, and salinization, thereby providing early warning and forecasting of major ecological disasters.

4. Furthermore, the natural vegetation on both sides of the TRB, such as *Populus euphratica* forests and desert shrubs, covers a relatively minor landscape area [46–48]. However, it plays a key role in supporting the landscape ecological processes and functions and would be affected severely if these vital landscape elements changed [49–51]. Meanwhile, it is the only barrier between the economic belt in the artificial oasis and the Taklimakan Desert [52,53]. Hence, ecological restoration of the TRlr will be a

long-term and gradual process. Such monitoring and predictions of TRlr can improve ecological security, provide scientific guidance for the optimal dispatch of water resources, and enhance ecological water use plans. Despite promising results, the performance of predictions of LUCCs and landscape pattern analysis in the future remains unclear as the influencing factors, such as climate, human management, and government policies, remain unchanged. Therefore, further studies are needed to analyze these factors more deeply and comprehensively and to take them into account in future predictions. This may include examining perspectives related to climate change, land management, ecological protection policies, and other relevant factors.

## 6. Conclusions

Based on remote sensing data for 2000–2020 and data simulated for 2030 using the CA–Markov model, the LUCCs and landscape pattern changes in the TRlr were analyzed and the impacts of those changes on the environment were discussed. In the TRlr, compared to the LUCCs of 2020, the LUCCs of 2030 predicted that a total of 3400 $km^2$ (20.29%) of the total area (16,760.94 $km^2$) would undergo changes. The areas of wetland, artificial land, grassland, farmland, and forestland were predicted to increase by 578.59, 43.90, 339.90, 201.62, and 536.11 $km^2$ (74.20, 51.31, 48.68, 23.30, and 10.55%), respectively. Only the areas of the Gobi/other deserts and bare soils (both classified under other land) were predicted to decrease by 1,700.13 $km^2$, (18.38%).

To determine the evolutionary characteristics of the TRlr landscape pattern indices at the class and landscape levels, we analyzed the trends in NP, PD, and AREA_MN for 2000–2020 and 2030 at the class level. The number and density of various land use patches decreased, and the mean patch sizes of the remaining land uses increased, except for other land. At the landscape level, our predictions showed that in 2030, LPI would increase, CONTAG values would increase compared to those in 2020, and SHDI and SHEI values would decrease compared to those in 2020. These landscape pattern indices were used to analyze the study area and reveal the characteristic variations between patches.

The findings of this study can serve as a basis for proposing several corresponding measures. Furthermore, we provided a discussion on the relationships between land development and the ecological environment in the context of efficient socioeconomic development and through scientific analyses, including the spatiotemporal evolution of land use patterns and land structure changes. The results of this study can promote coordinated and environmentally sustainable socioeconomic development within the TRB.

**Author Contributions:** Formal analysis, S.W.; funding acquisition, Q.Z. and K.Z.; investigation, J.W. and W.W.; methodology, S.W., Q.Z. and K.Z.; software, J.W. and W.W.; supervision, Q.Z.; writing—original draft, S.W.; writing—review and editing, S.W. and Q.Z. All authors have read and agreed to the published version of the manuscript.

**Funding:** This research was funded by the Key Project of National Natural Science Foundation of China-Xinjiang Joint Fund (U1803241), sponsored by the Natural Science Foundation of Xinjiang Uygur Autonomous Region (2022D01E095), the Chinese Academy of Sciences, "Western Young Scholars" Fund (2021-XBQNXZ-011), and Sub-project of the Third Xinjiang Comprehensive Scientific Expedition Project (2022xjkk1306-2).

**Institutional Review Board Statement:** Not applicable.

**Informed Consent Statement:** Not applicable.

**Data Availability Statement:** The data presented in this study are available on request from the corresponding author.

**Conflicts of Interest:** The authors declare no conflict of interest.

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
