# Peer review of "Predictions of Land Use/Land Cover Change and Landscape Pattern Analysis in the Lower Reaches of the Tarim River, China"

_land, doi:10.3390/land12051093_

Round 1

Reviewer 1 Report

Thanks for all your work, this research is interesting and comprehensive. The relationship between humans and land is an important aspect of geographical research. The authors discussed the temporal and spatial changes of land use/land cover change (LUCC) and landscape pattern evolution in TRlr since 2000 and 2020. The article uses the cellular automata Markov model and geographic information system analysis to simulate these changes over the 30 years of 2020 using 20 years of data. I recommend publishing this research in Land. To make the paper complete, here are a few questions and friendly suggestions. The details are as follows.

1. The Chinese expression mode of the article structure is very obvious, much like the writing method of a Chinese article. This requires the author to seriously consider refactoring. For example, (lines 38-39) In addition to being critical for economic growth and expansion across Central and South Asia, the TRB represents a strategic energy and resource reserve base supporting Chinas advancement. (line 75): Research prediction and simulation studies have also been conducted in the TRB.

2. The content of the introduction is obviously insufficient, and the research content is separated. The lines (58-80) lacks inductive and critical comments, and is mostly a pile of research. For example, Severe situations of increasing resource constraints, environmental pollution, and ecosystem degradation can directly threaten the sustainable development of the regional economy, society, and ecological security; hence, methods to improve the fragile TRB environment are vital[1-2]. I strongly recommend the author to add and refer to the following references.

[1] https://doi.org/10.3390/land11081303;

[2] https://doi.org/10.1016/j.jenvman.2022.115130

3. In addition, any research is carried out on the basis of predecessors, and the article neglects to use up-to-date references as research support. The author can start with the relevant research methods or content, and finally highlight the significance of this study, rather than simply similar research.

4.Some sentences are so long that it is hard to understand the sentences, I recommend that authors check the full text and refine the sentences.

5.In the section of discussion, the prospect of the article is not deep enough, please provide future research directions with the limitations of this paper. In addition, some of the discussion contents is redundant and needs to be refined.

Good luck!

Reviewer 2 Report

1. Give some models in applications. In other way we have to trust in your findings.

2. There is some contradiction in the paper - in rows 227 and 228 you said that artifitial lands scattered  in the middle and lower reaches was predicted to deccrease. In all ater your findings rgey are oredicted to increase. 

Round 2

Reviewer 1 Report

The article has greatly improved, and I recommend publishing it.